# Identification and Characterization of VDAC Family in Maize

**DOI:** 10.3390/plants12132542

**Published:** 2023-07-04

**Authors:** Carolina Rodríguez-Saavedra, Donají Azucena García-Ortiz, Andrés Burgos-Palacios, Luis Enrique Morgado-Martínez, Beatriz King-Díaz, Ángel Arturo Guevara-García, Sobeida Sánchez-Nieto

**Affiliations:** 1Laboratorio de Transporte y Percepción de Azúcares en Plantas, Departamento de Bioquímica, Facultad de Química, Universidad Nacional Autónoma de México, Ciudad de México C.P. 04510, Mexico; carol_unam10@hotmail.com (C.R.-S.); 418014145@quimica.unam.mx (D.A.G.-O.); aburgosbp@gmail.com (A.B.-P.); enrique.mgdomtz@gmail.com (L.E.M.-M.); kingbeat@unam.mx (B.K.-D.); 2Departamento de Biología Molecular de Plantas, Instituto de Biotecnología, Universidad Nacional Autónoma de México, Cuernavaca C.P. 62209, Mexico; arturo.guevara@ibt.unam.mx

**Keywords:** voltage-dependent anion channel, seedling development salicylic acid, methyl jasmonate, gibberellic acid, *Fusarium verticillioides*

## Abstract

The voltage-dependent anion channel (VDAC) is the most abundant protein in the outer mitochondrial membrane (OMM) of all eukaryotes, having an important role in the communication between mitochondria and cytosol. The plant VDAC family consists of a wide variety of members that may participate in cell responses to several environmental stresses. However, there is no experimental information about the members comprising the maize VDAC (ZmVDAC) family. In this study, the *ZmVDAC* family was identified, and described, and its gene transcription profile was explored during the first six days of germination and under different biotic stress stimuli. Nine members were proposed as *bona fide* VDAC genes with a high potential to code functional VDAC proteins. Each member of the *ZmVDAC* family was characterized in silico, and nomenclature was proposed according to phylogenetic relationships. Transcript levels in coleoptiles showed a different pattern of expression for each *ZmVDAC* gene, suggesting specific roles for each one during seedling development. This expression profile changed under *Fusarium verticillioides* infection and salicylic acid, methyl jasmonate, and gibberellic acid treatments, suggesting no redundancy for the nine *ZmVDAC* genes and, thus, probably specific and diverse functions according to plant needs and environmental conditions. Nevertheless, *ZmVDAC4b* was significantly upregulated upon biotic stress signals, suggesting this gene’s potential role during the biotic stress response.

## 1. Introduction

The voltage-dependent anion channel (VDAC) is the most abundant protein in the outer mitochondrial membrane (OMM) of all eukaryotes. It is a protein with a molecular weight of approximately 30 kDa with a highly conserved β-barrel structure, conformed by 19 β-sheets with an antiparallel order, and a single α-helix at its N-terminal region, which is located inside the hydrophilic pore [1,2]. The VDAC has several functions, such as being an exchange channel for key metabolites and ions between mitochondria and cytosol in a voltage-dependent manner, and it also regulates mitochondrial and cellular energy conversion and influences membrane permeability by participating in the formation of mitochondrial permeability transition pore (mPTP) [3]. Due to these activities, the VDAC has an important role in the communication between mitochondria and cytosol, participating in stress responses through several signal transduction mechanisms that are activated via partner binding [2,4,5].

In mammals, the VDAC gene family consists of three isoforms, in which hVDAC1 has been extensively studied due to its ability to interact with different partners to promote several gating conformations. Altered VDAC protein interactions can lead to the development of diseases, such as cancer and neurodegenerative disorders [2,6]. In plants, the VDAC family consists of a larger variety of members: *Nicotiana benthamiana* and *Nicotiana tabacum* have three isoforms [7]; *A. thaliana* [8] and *Lotus japonicus* [9] have five VDAC genes; *Vitis vinifera* and *Vitis piasezkii* have six [10], *Oryza sativa* has eight [11], *Populus tricocarpa* has ten [1], and *Triticum aestivum* has twenty-six [12]. This multiplicity of VDAC isoforms in plants suggests a wider variety of functions than those found in mammals. Indeed, plant VDACs are involved in growth, development, and reproduction [8], as well as in several environmental stresses. For example, in *Triticum aestivum*, VDAC1 (TaVDAC1) confers tolerance to salt stress but less resistance to drought [12], while *Vitis piasezkii* VDAC3 (VpVDAC3) plays a role in plant resistance to *Plasmopora viticola* infection [10]. *Arabidopsis thaliana* VDAC1 (AtVDAC1) is involved in the defense response against avirulent pathogens [8], and AtVDAC2 has a role in the signal transduction pathway under salt stress [5]. VDACs have also been suggested to act as direct regulators of programmed cell death (PCD): in *N. tabacum*, VDAC3 (NtVDAC3) is involved in Bax-induced cell death [7], and overexpression of *O. sativa* VDAC4 (OsVDAC4) in *N. benthamiana* leaves triggers PCD, while its co-expression with hexokinase 3 from *N. tabacum* (NtHXK3) mitigates this process [13].

All these findings suggest that plant VDACs participate in cell responses to several environmental stresses, probably implicating hormone signaling, such as salicylic acid (SA) that induces activation of pathogen-related proteins (PRs) and hypersensitive response (HR), which in turn activates PCD [5,14]. However, despite the experimental findings supporting these suggestions, plant VDACs’ response to SA and other hormone signaling pathways in response to biotic stresses has been poorly explored.

Biotic stress affects plant productivity worldwide. For example, in maize, it contributes to an average of 22.5% in crop losses [15], with a high impact on food security, as maize provides about 42% of the world’s food calories in addition to its use as a livestock feed crop, industrial raw material, and energy crop [16]. Understanding the maize responses to pathogen attacks is an important problem to solve. VDAC family in maize (*ZmVDACs*) could be affected by pathogen infection; however, apart from transcriptomic data [17,18], experimental evidence of VDAC is scarce. The presence of VDAC activity was detected in maize plastidial membranes [19], and conductance measurements were made in a purified protein from mitochondria [20], but there is no additional experimental information about the members comprising the family and their characteristics.

In this study, the ZmVDAC family sequence features, protein tertiary structures, and phylogenetic relationships were analyzed to propose nomenclature for each member of the family. Additionally, an analysis of their transcriptional level profile during the early days of germination and in response to *F*. *verticillium* infection was conducted. The effect on *ZmVDACs* transcription levels when seeds were treated with gibberellic acid (GA), SA, and methyl-jasmonate (MeJa) as signaling hormones of the biotic stress response were also investigated.

## 2. Results

### 2.1. Identification of VDAC Gene Family Members in Maize

To identify the sequences that code VDAC proteins in maize, a standard database screening on UniProt [21] was conducted using the BLAST tool and the maize protein sequence A0A317Y8M0_MAIZE (UniProt) as a query. This sequence was annotated as a mitochondrial outer membrane protein porin from maize. A total of 38 sequences were originally obtained, but after an additional search on NCBI [22], ENA [23], Ensembl Plants [24], and Maize GDB [25] databases, this number was reduced to 18 sequences. They ranged from 828 to 1275 nucleotides with the translated sequences consisting of 275 to 424 amino acids, with a predicted molecular weight between 27.39 to 45.00 kDa (Appendix A). The gene structure of most of these 18 sequences was similar; they had 6 exons and 5 introns, but UTRs were identified only in 10 sequences: ZmVDAC1a, ZmVDAC1b, ZmVDAC2_2, ZmPorin1_0, ZmVDAC3_1, ZmUP, ZmVDAC4, ZmOMMPP, ZmMOMPP2, and ZmOPMPPPOR1 (Appendix A).

A closer examination of ZmOPMPPPOR1 and ZmPorin1_0 sequences showed that both sequences had a localization almost in the same locus on chromosome 8 (Appendix A), with ZmOPMPPPOR1 having an additional two repeated GTG nucleotide at the 5′ end of the sequence, which gives the translated sequence two additional valines that are not present in the other porin sequences (Appendix A). Hence, the sequence ZmOPMPPPOR1 was excluded from the later analysis. Therefore, nine sequences were considered putative members of the maize VDAC family.

### 2.2. Mitochondrial Porin Signature and Putative 3D Structures of ZmVDAC Family Members

As part of sequence characterization and description of structure from VDAC members, mitochondrial porin signature (MPS) was localized in 17 ZmVDAC sequences. MPS is a C-terminal domain composed of 23 amino acids, conserved in all VDAC proteins, whose canonical sequence is as follows: [YH]-X(2)-D-[SPCAD]-X-[STA]-X(3)-[TAG]-[KR]-[LIVMF]-[DNSTA]-[DNS]-X(4)-[GSTAN]-[LIVMA]-X-[LIVMY], where X represents any amino acid [8]. In plants, MPS has been classified into two groups: the canonical or conserved MPS motif and the divergent motif [7]. A multiple sequence alignment of the putative ZmVDAC proteins with known VDAC plant families showed that the MPS sequence was present in all the orthologues analyzed (Appendix A), with some of them, such as AtVDAC1, AtVDAC2, AtVDAC3, AtVDAC4, AtVDAC5, VvVDAC1, VvVDAC3, VvVDAC5, and VvVDAC6, carrying both the conserved and the divergent MPS motif [8,10]. The MPS domain was found in ZmVDAC1a, ZmVDAC1b, ZmVDAC2_2, ZmPorin1_0, ZmVDAC3_1, ZmUP, ZmVDAC4, ZmOMMPP, ZmMOMPP2, ZmOMM, and ZmVDAC3_0 around amino acids 219 and 241; for ZmMOMPP4, MPS was found at amino acid 226, at 246 for ZmPorin1_1, at 257 for ZmVDAC2_1, at 247 for ZmVDAC2_0, at 209 for ZmOMMPorin1, and at 367 for ZmVDAC6 (Appendix A). The alignment showed that most of the ZmVDACs had the conserved MPS domain, except for ZmUP, ZmVDAC4, ZmOMM, ZmMOMPP4, ZmVDAC3_0, and ZmVDAC2_1 that had divergent MPS domains.

Another approach to determining the possibility of the sequence coding for functional VDAC proteins was by predicting their tertiary structure. Sixteen sequences presented at least one α-helix at the N-terminal region and 19 β-sheets in a β-barrel conformation, while only the ZmVDAC6 sequence showed a spiral conformation (Figure 1a). Eleven structures—ZmVDAC1a, ZmVDAC1b, ZmVDAC2_2, ZmPorin1_0, ZmVDAC3_1, ZmUP, ZmVDAC4, ZmOMMPP, ZmMOMPP2, ZmOMM, and ZmMOMPP4—had a high prediction confidence score, as noted by the pLDDT value > 90, while the rest had pLDDT values < 90, indicating low prediction reliability for these structures.

Additionally, ZmVDAC 3D predicted structures were superimposed with the murine VDAC1 (MmVDAC1) [26] to determine the similarity between the proposed maize VDAC sequences and a well-known functional VDAC protein whose structure has already been solved. Figure 1b shows a high similarity between MmVDAC1 and ZmVDAC1a, ZmVDAC1b, ZmVDAC2_2, ZmPorin1_0, ZmVDAC3_1, ZmUP, ZmVDAC4, ZmOMMPP, ZmMOMPP2, ZmOMM, ZmVDAC3_0, ZmVDAC2_1, and ZmPorin1_1, given root mean square deviation (RSMD) values that are similar to or lower than the values of the superimposed structures of AtVDAC1 and MmVDAC1 (2.49 Å), thus providing high confidence in the similarity of the sequences (Figure 1b).

Together, pLDDT (>90) and RSMD values (2.5–2.2 Å) suggested that the sequences with the highest probability and reliability of generating functional VDAC proteins were ZmVDAC1a, ZmVDAC1b, ZmVDAC2_2, ZmPorin1_0, ZmVDAC3_1, ZmUP, ZmVDAC4, ZmOMMPP, ZmMOMPP2, and ZmOMM. Of these 10 listed sequences, ZmOMM was no longer analyzed since no UTR regions were found for this sequence (Appendix A).

### 2.3. Phylogenetic Analysis and Defining a Homogeneous Nomenclature for ZmVDAC Family Members

To explore the evolutionary relationship of the nine chosen ZmVDAC protein sequences among their orthologues in other plants, a phylogenetic tree of four monocotyledonous plants (maize, wheat, rice, and pearl millet) and four dicotyledonous plants (Arabidopsis, tobacco, grapevine, and *L. japonicus*) was constructed, including MmVDAC1 to root the tree. Forty-one plant VDAC sequences were clustered into four clades, and ZmVDACs were found in groups A, B, and D (Figure 2), away from dicots but closer to other monocots, like wheat and rice; however, they all gathered in different groups according to their sequence diversity, suggesting the functional divergence of ZmVDAC proteins. Groups A and B clustered sequences of both monocots and dicots, while groups C and D exclusively clustered respective VDAC sequences from dicots and monocots.

From this cladogram, nomenclature for ZmVDACs was proposed according to their proximity to the orthologue whose protein expression or activity was already reported. ZmUP and ZmVDAC4 were grouped in the same clade (A) as AtVDAC4 [8] so they were renamed as ZmVDAC4a and ZmVDAC4b, respectively. ZmOMMPP and ZmMOMPP2 in clade B were close to OsVDAC5 [8], so they were renamed as ZmVDAC5a and ZmVDAC5b, respectively. Finally, ZmVDAC1a, ZmVDAC1b, ZmVDAC2_2, ZmPorin1_0, and ZmVDAC3_1 were clustered in clade D, with TaVDAC1 [12] and OsVDAC1 [27] closer to ZmVDAC1a and ZmVDAC1b; hence, these two sequences kept their original annotation. ZmVDAC2_2 was close to OsVDAC2 [27] and TaVDAC2 [12], so it was renamed as ZmVDAC2. Finally, ZmPorin1_0 and ZmVDAC3_1, which were close to OsVDAC3 [27] and TaVDAC3 [12], were renamed ZmVDAC3a and ZmVDAC3b, respectively (Table 1).

### 2.4. Tissue-Specific Expression of ZmVDAC Genes According to RNA-Seq Database

Transcript levels of the nine maize VDAC genes proposed as the most likely to encode functional proteins were found at the eFP Atlas Browser at Maize GDB, confirming that the nine sequences proposed are the ones that have a higher potential to generate functional products.

Figure 3 shows that the only two genes that are at very low levels in almost all tissues and treatments are *ZmVDAC3b* and *ZmVDAC4a*. *ZmVDAC3a*, *ZmVDAC4b*, *ZmVDAC5a*, and *ZmDAC5b* are moderately expressed in all tissues, while *ZmVDAC1a*, *ZmVDAC1b*, and *ZmVDAC2* are mainly expressed during the early stages of seed embryogenesis and seedling development. *C. gramminicola* or *C. zeina* infection increases the expression of *ZmVDAC1a*, *ZmVDAC1b*, and *ZmVDAC2*. The most abundant transcript in the leaves under *C. zeina* infection is *ZmVDAC1a*. On the contrary, *ZmVDAC5a* decreases under *C. zeina* infection. These data confirm that the selected genes are transcribed and suggest tissue- and condition-specific expression of *ZmVDAC* genes, which needs to be explored further.

### 2.5. Expression Profiles of ZmVDACs Genes in Coleoptiles during the First 6 Days of Germination

To obtain the transcript level profiles for the *ZmVDAC* family in coleoptiles during the first 6 days of germination, qPCR analysis was conducted using the transcript level of *ZmVDAC* genes in embryos at 0 h of germination as the control (Figure 4). However, it was not possible to follow transcript level profiles for *ZmVDAC3a*, *ZmVDAC5a*, and *ZmVDAC5b* because designed primers for the last two transcripts were not specific enough for each sequence (Appendix A), given the high identity (96%) between them (Appendix A). In the case of *ZmVDAC3a*, it was not possible to obtain specific primers either (Appendix A) even though different primers were tested. Nevertheless, the amplification product for *ZmVDAC5a* and *ZmVDAC5b* was detected after the standard PCR in agarose gel electrophoresis (Appendix A). *ZmVDAC5b* expression levels were low during germination while *ZmVDAC5a* was detected at 48 h; then it slightly decreased with time. No product was observed on agarose gel electrophoresis for *ZmVDAC3a* after the standard PCR (Appendix A).

qPCR analysis showed that, relative to control levels (0 h of germination), *ZmVDAC1a, ZmVDAC2, ZmVDAC3b*, and *ZmVDAC4b* were downregulated on day 1, while *ZmVDAC1b* and *ZmVDAC4a* remained at control basal levels. However, on day 2, *ZmVDAC1a*, *ZmDAC1b*, and *ZmDAC4a* were upregulated almost 10 times, while *ZmVDAC2*, *ZmVDAC3b*, and *ZmDAC4b* remained downregulated. This trend was practically maintained on day 3, but on day 6, *ZmVDAC* genes were all again downregulated as on day 1 (Figure 4). These results suggest specific functions for each *VDAC* gene during germination and seedling development.

### 2.6. Identification of Cis-Acting Regulatory Elements in the Promoter Region of ZmVDAC Genes

*Cis*-acting regulatory elements of genes are particularly important to promoting or inhibiting gene expression under certain specific conditions. The nine *ZmVDAC* genes displayed response elements to different stimuli related to biotic stress: two regulatory response elements to elicitors were found, one of them exclusively found for *ZmVDAC1a*, and the second one found in all *ZmVDAC* genes, except for *ZmVDAC1b* and *ZmVDAC3a* (Figure 5). Additionally, a diversity of regulatory response elements to SA, JA/Et, and GA were found for all nine *ZmVDAC* genes, with a different number of repeats. These predictions suggest that the *ZmVDAC* family may have specific functions under biotic stress responses.

### 2.7. Morphological Comparison between Healthy and Infected Seedlings with Fusarium verticillioides

*F. verticillioides* produces diseases in several economically important crops all over the world [29,30]. In maize, *F. verticillioides* infection is commonly detected only when maize has reached maturity; however, invasion and infection can start from fungal conidia that are either carried inside the seeds or on the seed surface. Hence, the fungus develops inside seedlings, moving from the roots to the stalk and finally to the cob and kernels [31].

In this study, to determine the early events of *F. verticillioides* seedling infection and its effect on the transcriptional profile of the *ZmVDAC* family along with their possible role in biotic stress response, maize seeds were primed with *F. verticillioides* for 6 days (dpi). Seeds without priming were used to obtain healthy seedlings (control). Figure 6a shows no visible effects of *F. verticillioides* infection on seedlings during 6 dpi, as primary roots and coleoptile growth were very similar between control and infected seedlings during the first 3 days of germination, even though a clear invasion of the fungus in the seedlings was easily noticed (Appendix A). Morphological observations are consistent with the lack of differences between healthy and infected primary roots and coleoptile size (Figure 6b). However, on day 6, despite the larger primary roots of the healthy seedlings, the number of secondary roots was notably higher in the infected seedlings (Figure 6c). No sign of cell death nor other symptom of infection was observed.

### 2.8. Maize Seedlings Response to F. verticillioides Infection

To verify the presence of *F*. *verticillioides* on the seedling, healthy and infected seedlings were homogenized at different germination times, and serial dilutions of the samples were loaded into PDA plates. At early times of germination (1 day), all infected seedlings presented around 1000 colony-forming units (CFU), reaching almost 10 million CFU at 6 dpi, while there was no fungus growth in control seedlings (Appendix A).

Pathogen infection often produces several responses and damaged areas in host cells, including loss of membrane potential, an increase in cellular respiration, a burst of reactive oxygen species (ROS), and localized cellular death [10,32,33]. To test if *F. verticillioides* caused membrane damage, changes in seedling viability, or overproduction of ROS during the first 6 days of infection, electrolyte leakage (Figure 7a), and H_2_O_2_ production (Figure 7b) were monitored, and a cell viability staining was performed at 6 dpi to verify cell death (Figure 7c). Electrolyte leakage increased with germination time in both healthy and infected seedlings, showing no significant differences between the two conditions (Figure 7a). Control seedlings reduced their H_2_O_2_ production with germination time; on the contrary, the infected maize seedlings, in a time-dependent manner, accumulated H_2_O_2_, indicating that maize seedlings were sensing *F. verticillioides* infection (Figure 7b). This result was further confirmed with a histochemical 3,3′-Diaminobenzidine (DAB) staining of cotyledons from 6 days-old seedlings (Figure 7c). A reddish-brown stain was formed by the reaction of DAB with the endogenous H_2_O_2_, allowing for qualitative visualization of H_2_O_2_ produced by the plant tissue. The staining was stronger in infected cotyledons than in healthy, demonstrating that seedlings were perceiving *F. verticillioides* invasion.

Evan’s blue staining was conducted to visualize possible cell death on 6-day cotyledons (Figure 7c), as Evan’s blue is a vital stain that selectively accumulates in dead cells due to membrane damage. Blue staining was observed for both treatments, with no evident difference in infected cotyledons. This was probably due to natural cell death occurring in cotyledon as a tissue that only promotes and sustains the proliferation of true leaves. Thus, no cellular death was observed due to pathogen infection.

Together, all these results confirmed that seedlings’ infection with *F. verticillioides* was indeed asymptomatic, such that occurs in harvest fields.

### 2.9. Expression Profiles of ZmVDACs Genes under Biotic Stress Signals

#### 2.9.1. Expression Profiles of Defense Genes and ZmVDAC Genes during *F. verticillioides* Infection

Even though the aerial part of the seedlings did not show any evidence of disease, it was still possible that a molecular response from maize seedlings was generated due to *F. verticillioides* infection. To explore this possibility, some genes associated with defense response to biotic stress (*ZmPR1*, *ZmPDF1*, and *ZmMCAS6*) were analyzed by qPCR to determine if they were activated. Figure 8a shows the transcript level graph for these genes in the base 10 logarithm, using cDNA from uninfected coleoptiles as the control. The result showed that during the 6 days of germination under healthy conditions, transcript levels of defense genes remained at basal levels; however, upon *F. verticillioides* infection, *ZmPR1* transcripts significantly increased on day 1 (1000-fold increase) and day 6 (100-fold increase), while on days 2 and 3, there were no differences regarding the healthy condition, probably due to fungus evasion of seedling defense response. On the other hand, *ZmPDF1* was only significantly upregulated on day 1, increasing almost 100 times compared with basal levels. Finally, though *ZmMCAS6* levels slightly increased on the first, third, and sixth days of infection, but these increases were not significant (Figure 8a).

To know if *F. verticillioides* induced changes in the transcript levels of *VDAC* genes, a qPCR analysis using cDNA from infected coleoptiles was conducted. Figure 8b shows that the *ZmVDAC* transcript level profile was different from that obtained during germination under healthy conditions (Figure 4): on day 1, all six analyzed genes were upregulated, including *ZmVDAC4b*, which augmented its transcript level up to 30 times relative to the healthy condition, as well as *ZmVDAC2* and *ZmVDAC3b*, even though their increase was slight. *ZmVDAC1a* increased its levels up to 100 times compared with healthy coleoptiles (Figure 8b). Infection on day 2 downregulated the expression of *ZmVDAC1b*, *ZmVDAC2*, *ZmVDAC3b*, *ZmVDAC4a*, and *ZmVDAC4b*, and only *ZmVDAC1a* maintained a similar transcript level as the control, while on day 3 *ZmVDAC2*, *ZmVDAC3b ZmVDAC4a*, and *ZmVDAC4b* remained downregulated. This time, only *ZmVDAC1b* kept a similar transcript level as under the healthy conditions, while *ZmVDAC1a* was downregulated. Finally, on day 6, all six *ZmVDAC* genes increased their transcript level contrary to the healthy condition: *ZmVDAC1a* and *ZmVDC1b* augmented their levels up to 10 times, *ZmVDAC2* increased 5 times, *ZmVDAC4a* and *ZmVDAC4b* slightly increased up to 2 times, and *ZmVDAC3b* remained at the same level as the control. In brief, these results indicate that infection with *F. verticillioides* affects the *ZmVDAC* expression profile.

#### 2.9.2. Effect of Salicylic Acid, Methyl Jasmonate, and Gibberellic Acid on Transcript Level Profile of ZmVDAC Genes in 72 h Coleoptiles

To explore if the *ZmVDAC* family is regulated by hormones implicated in biotic stress response, assays treating the seeds with SA, MeJa, and GA were performed. GA levels were monitored during the first 50 h of *Fusarium*-infected maize seeds, and it was found that GA levels were higher on the infected seeds than on the healthy ones (Appendix A). Given this, GA was also chosen as a hormone probably involved in some type of response to *Fusarium* invasion.

Expression profile of the six *ZmVDAC* genes in coleoptiles at 3 d of germination after SA treatment shows that *ZmVDAC1a*, *ZmVDAC1b*, *ZmDAC4a*, and *ZmVDAC4b* were upregulated compared with control coleoptiles, with *ZmVDAC1a* and *ZmVDAC4b* showing the highest transcript level, while *ZmVDAC2* and *ZmVDAC3b* were downregulated (Figure 8d). On the other hand, despite the retarded growth of seedlings treated with MeJa, expression of *ZmVDAC1b* and *ZmVDAC4b* genes was higher than in the control, with an abundance similar to that found in SA treatment. *ZmVDAC1a* and *ZmVDAC4a* remained as the control, and *ZmVDAC2* and *ZmVDAC3b* were kept downregulated, as with SA treatment. GA treatment caused downregulation of *ZmVDAC1a, ZmDAC2*, and *ZmVDAC4a* below control levels, contrary to *ZmVDAC1b*, which was upregulated but similar to SA and MeJa treatments; *ZmVDAC3b* was slightly activated upon GA treatment. *ZmVDAC4b* was the most hormone-concentration-sensitive gene.

## 3. Discussion

VDAC proteins have drawn extensive attention since their location at the OMM and transport function are key to mediating cross-communication between mitochondria and cytoplasm to regulate multiple cellular processes [14]. In plants, the VDAC protein family has larger members than in mammals, and yet all their possible functions are just beginning to be explored. As sessile organisms, plants are subjected to a variety of environmental stresses that may lead to reduced or even total loss of crop production [15]. Therefore, it is important to analyze the proteins and mechanisms that plants use to cope with environmental stresses. According to experimental evidence, members of the VDAC family in wheat, Arabidopsis, *N. benthamiana*, and grape participate in vital processes such as programmed cell death, resistance to pathogen infection, salinity, cold, and drought stresses [7,12,13,34]. In maize, though there is some data about VDAC transcripts level [17,18] as well as experimental evidence of the presence of VDAC activity in maize plastidial and mitochondrial membranes [19,20,35], there is a need for a more exhaustive characterization. For example, defining the family members will help to test its transcriptional regulation, or to determine the specific activity of each member to provide insights about its physiological function in the plant.

In this study, a total of nine ZmVDAC sequences were selected after database screening and different in silico analyses, such as (a) a thorough analysis of all nucleotide and protein databases, to avoid misannotated or redundant sequences; (b) gene structure analysis to find exons, introns, and UTRs, to select sequences with the highest probability of been transcribed and/or translated; (c) inference of β-barrel 3D structure, to know if sequences have a high probability to form a canonical VDAC structure, similar to those already explored and characterized; and (d) determination of their phylogenetic relationships. All these filters were applied to select the sequences with the highest probability to generate functional VDAC proteins and served to suggest nomenclature for the maize family proposed as ZmVDAC1a, ZmVDAC1b, ZmVDAC2, ZmVDAC3a, ZmVDAC3b, ZmVDAC4a, ZmVDAC4b, ZmVDAC5a, and ZmVDAC5b (Table 1).

The nine ZmVDAC selected sequences had a wide range of identity among them, from 39.6 to 96.4% (Appendix A); nevertheless, secondary and tertiary structures were quite similar: they all contained 19 β-sheets and a single α-helix at the N-terminal region, and they all exhibited an MPS motif near the C-terminal region (Appendix A). Three-dimensional structures also had similar conformations among them and among other VDAC families (Figure 1a), as has been shown for all VDAC families studied so far [10,12,36]. The protein 3D structure conservation suggests that nine selected ZmVDAC sequences may conserve functions as well, due to their high similarity to MmVDAC1 (Figure 1b).

Despite these structural similarities, some slight variation in the MPS domain could drive an important alteration in function. Within the nine proposed members of the ZmVDAC family, seven of them contained a conserved MPS motif and the remaining two contained a divergent MPS domain (Appendix A): ZmVDAC4a and ZmVDAC4b, both with a cysteine at amino acid position 241 and ZmVDAC4b additionally with a noncanonical methionine at amino acid 231. Tateda et al. [8] found that AtVDAC2 and AtVDAC4 had divergent MPS domains due to a single change at amino acid position 223, which was found sufficient to localize these two proteins not only in OMM but also in cell peripheral vesicles. However, different MPS motives were found for ZmVDAC sequences, as well as variants have been found for *Lotus japonicus*, grape, or wheat VDACs [9,10,11]. These observations show that different types of divergent MPS motifs exist within plant VDAC protein members from the same species, suggesting that emerging mutations may produce nonredundant isoforms, probably with additional functions. However, MPS domain variants need to be further explored in order to determine if they could potentially re-localize VDACs to any other subcellular compartment, or if they could have additional functions. Phylogenetic analysis also shows the divergence of each sequence within the same species, suggesting that there is no redundancy between VDAC isoforms. Five of the ZmVDAC sequences, ZmVDAC1a, ZmVDAC1b, ZmVDAC2, ZmDAC3a, and ZmVDAC3b, were clustered within a clade exclusively defined for monocots that may suggest conserved functions. The remaining four, ZmVDAC4a, ZmVDAC4b, ZmVDAC5a, and ZmVDAC5b, were grouped in clades that show evolutionary divergence for both dicots and monocots (Figure 2), which may be a predictor of additional functions that have yet to be identified. Moreover, far beyond these observations, the phylogenetic analysis allowed us to propose an adequate nomenclature for each member of the ZmVDAC family, according to previously identified and characterized orthologues.

As part of the identification of the ZmVDAC family, transcriptomic data for each member was confirmed by databases [17,18]. eFP Atlas Browser reports the existence of transcripts for the nine proposed members of the *ZmVDAC* family at different stages of maize development (Figure 3), showing different transcript levels for each one, depending on tissue and stage of development, as seen for VDACs from *A. thaliana* [8], *V. vinifera* [10], and *T. aestivum* [12]. Data obtained from eFP Atlas Browser show important information about *ZmVDAC* transcript levels starting from 6 days after sowing; however, information about the early stages of germination was not available. Hence, in this study, transcript-level profiles of *ZmVDACs* in coleoptiles during the first 6 days of seedling development were obtained (Figure 4).

Seedling development affected the expression of the *ZmVDAC* family. The RNA levels on day 1 of germination were minimal, which may have been due to a delay in the transcription since functional mitochondria were detected upon seed imbibition. Mitochondrial proteins could be synthesized during embryogenesis leading to low amounts of transcripts [37]; however, an increase in the abundance of *ZmVDAC1a*, *ZmVDAC1b*, and *ZmVDAC4a* transcripts on days 2 and 3 could be explained by the emergence of new tissues. Later, on day 6, seedlings had similar levels of *ZmVDAC* transcripts to the first day of germination. The decrease on day 6 did not entirely match the data reported by Stelpflug et al. [17] and Hoopes et al. [18], who found a high expression level for *ZmVDAC1a*, *ZmVDAC1b*, and *ZmVDAC2*, and a medium expression level for *ZmVDAC3a*, *ZmVDAC5a*, and *ZmVDAC5b* (Figure 3). In this study, *ZmVDAC1a* and *ZmVDAC2* were downregulated, while *ZmVDAC1b* and *ZmVDAC4a* remained as the control. These differences may be due to different seed germination conditions, such as sowing medium and incubation. Nevertheless, these findings suggest that each member of the *ZmVDAC* family has a specific role and expression during seed germination and seedling development; interestingly, it seems that out of all six transcripts evaluated, *ZmVDAC2*, *ZmVDAC3b*, and *ZmVDAC4b* were not required during germination under the explored conditions. This expression profile, in which all transcripts are not upregulated together, has been reported for other plant VDAC genes, and it may suggest a way to avoid redundancy [34]. In the case of VDACs in mammals, it has been stated that each isoform can affect the expression of others to maintain coordination [38]. Additionally, these results agree with other plant VDAC families, in which it has been found that VDAC expression is age- and tissue-dependent, such as VDACs from *A. thaliana* [8], grapevine [10], and wheat [12]. However, this is the first reported plant VDAC transcript tracing during the early growth development and is noticeable that transcript profiles are also time-dependent within the same tissue.

Apart from the spatiotemporal expression of plant VDACs, their expression may also vary depending on external or internal cues. Previous studies have found that protein VDACs among all eukaryotes are capable to interact with mitochondrial and cytoplasmic proteins to regulate several mitochondrial-related resistance responses [6]. Indeed, plant *VDAC* transcripts and proteins have been confirmed to have a role during a variety of stresses: *AtVDAC1* is essential to slow down colonization of *P. syringae* [8]; *N. tabacum* VDACs are involved in defense response against *P. cichorii* [7]; AtVDAC2 has a role in salt stress response pathway [39] as well as in salicylic acid signaling pathway [5]; VpVDAC3 enhances tolerance to *P. viticola* infection [10]; and TaVDAC1 has been found to confer tolerance to salinity stress [12].

In this study, the biotic stress response of *ZmVDAC* family was of particular interest due to the number of pathogenic species that affect maize production and quality and due to the efforts to elucidate defense-associated genes and proteins in food crops [40]. Analysis of *cis*-acting response elements predicted that the abundance of *ZmVDACs* members could be influenced by biotic stress signals, such as elicitors or SA and JA/ethylene hormone signaling (Figure 5), suggesting the involvement of this gene family in the maize biotic stress response. Interestingly, according to Stelpflug et al., [17] and Hoopes et al. [18], *ZmVDACs* expression is modified during seedling pathogen infection, as these authors showed a slightly higher abundance of *ZmVDACs* on maize leaves under *Colletotrichum graminicola* and *Cercospora zeina* infection (Figure 3). In this study, we focused on the effect of *F. verticillioides* infection on the *VDAC* expression profile because this fungus is one of the main causes of crop loss around the world. This way, asymptomatic seedlings with *F. verticillioides* infection (Figure 6 and Figure 7) were produced, which is a common condition in the field, as this fungus is an endemic and systemic pathogen for maize until growing conditions change. The fungus turns to necrotrophy, producing maize stalk and ear root diseases that are difficult to eradicate when maize reaches maturity [32].

However, even though seedlings did not show visible signs of defense response (Figure 6 and Figure 7a,c), very particular differences could be observed in seedling growth, as several secondary roots were significantly augmented in 6-day-infected seedlings in contrast to 6-day-healthy seedlings (Figure 6a,c). Oren et al. [30,31] had already reported that maize seeds inoculated with *F. verticillioides* did not repress seedling growth but promoted it. Lately, Zeng et al. [40,41] proposed that fumonisin B1 (FB1), a *Fusarium*-produced carcinogenic mycotoxin, was responsible for an alteration in auxin signaling pathway, necessary to promote the emergence of secondary roots. In this study, *Fusarium verticillioides* MY3, an FB1-producing strain, was used for all the infection assays; hence, there is a high probability that inoculum concentration and FB1 production by *F. verticillioides* MY3 at the conditions assayed were somehow adequate to promote an increase in secondary roots, as well as to produce asymptomatic seedlings. However, there may be another explanation for the early start of root growth that could work together with the previous suggestion: *Fusarium* production of gibberellic acid during maize embryo germination (Appendix A) disrupts the hormone balance in favor of promoting seed germination.

The increased production of ROS during *F. verticillioides* invasion (Figure 7b) showed that defense response was indeed displayed at a cellular and molecular level. Under pathogen invasion, ROS are rapidly produced as part of the first line of defense in plants, which eventually triggers the activation of downstream signaling networks, which culminates in the activation of several transcription factors that promote the transcription of defense genes, such as *PR1*, *PR10*, *PDF1*, or chitinases [32,42]. According to this, Figure 8a shows that transcript levels for *ZmPR1* were significantly increased on day 1 and day 6 after *Fusarium* inoculation, correlating with the days when seedlings presented the highest increase in ROS (Figure 7b). On the other hand, there were slightly more transcripts of *ZmMCAS6* on day 6. *PR1* is a gene that codes pathogenesis-related protein 1, one of the major response molecules generated upon the SA signaling pathway, which accumulates as a response to pathogen infection and systemic acquired resistance (SAR), providing tolerance to pathogen invasion by initiating HR [5,43]. MCAS6 is a member of the plant caspase-like proteases family, which has been related to HR cell death and PCD associated with other nonpathogenic responses [44]. The increase in these two transcripts on maize seedlings during *F. verticillioides* infection on the first and sixth days shows that seedlings perceived fungus invasion at these time points, suggesting that the SA signaling pathway was activated, though HR was not achieved, as levels of *ZmMCAS6* were not significantly increased compared with basal control levels, which was confirmed by Evan’s blue staining (Figure 7c). These results agree with Wang et al. [29] who reported that during *F. verticillioides* invasion on maize kernels, SA-induced genes were activated, along with those induced by ABA and JA signaling; however, Lanubile et al. [45] found that JA/Ethylene signaling pathway was activated and did not observe a significant induction of SA-related genes. This discrepancy in results from different studies has been attributed to differences in pathogenic strains, maize races, and stages of development, as well as to pathogen inoculation methods [29].

Activation of a defense response at the molecular level upon *F. verticillioides* seedlings infection anticipated changes in the *ZmVDAC* transcript profile. qPCR analysis proved that maize VDAC genes exhibited different transcript levels during pathogen invasion: *ZmDAC1a*, *ZmVDAC1b*, *ZmVDAC2*, *ZmVDAC3b*, *ZmVDAC4a*, *ZmVDAC4b*, and even *ZmVDAC5a* and *ZmVDAC5b* (Appendix A) were all affected upon *F. verticillioides* infection (Figure 8b). However, the most striking finding was that *ZmVDAC2* and *ZmVDAC4b* were significantly upregulated during infection, contrary to healthy conditions, in which these transcripts were apparently not required for seedling development (Figure 4), suggesting a potential role for *ZmVDAC4b* and *ZmVDAC2* during the biotic stress response, as seen in NtVDACs, AtVDACs, and VpVDAC3 [7,8,10]. Additionally, increased levels of all evaluated *ZmVDAC* genes correlated with ROS and *PR1* increases on the first and sixth days of *Fusarium* infection (Figure 7b and Figure 8a,b). The release of ROS during cell respiration is an event that could be controlled by VDAC, but ROS can also regulate the abundance of VDAC. The addition of methyl viologen, a strong oxidizing agent, to Arabidopsis plants induces *AtVDAC2* and *AtVDAC4* expression, whereas ROS content is reduced in the mitochondria when yeast mutants lacking ScVDAC1 (∆por1 mutant) are complemented with AtVDACs [34]. In the same context, overexpression of plant VDACs has proved to increase *PR1* levels [7,46] and conversely, under SA treatment, VDAC expression increases significantly [47]. Additionally, overexpression of AtVDACs conferred sensibility to SA, while AtVDAC null mutants were insensitive to this hormone, suggesting that VDACs could be participating in the SA signaling pathway [5]. Results in the present study may suggest the same hypothesis, as *ZmVDACs* genes were upregulated when ROS and *PR1*, two of the main players in the SA signaling pathway, were upregulated too. Nevertheless, several additional experiments are needed in maize to have sufficient evidence to confirm this hypothesis.

Since several hormones are modulators of the plant responses to pathogen infection, further exploration of *ZmVDAC* transcript levels was conducted by treating the seeds with SA, MeJa, and GA and then germinating for 3 days (Figure 8d). GA treatment was performed since *F. verticilioides* induces its accumulation in maize (Appendix A). *ZmVDAC* genes were regulated by all three hormones, as *cis*-acting regulatory elements predicted (Figure 5), resulting in a different transcript profile in coleoptiles regarding transcript pattern during germination (Figure 4) and during *F. verticillioides* infection (Figure 8b), suggesting specific roles for certain *VDAC* genes upon these three hormones treatments (Figure 8d). Noticeably, all hormones produced different phenotypes in seedling growth; however, most of the *ZmVDACs* were upregulated at similar levels. *ZmVDAC4b* was the most sensitive to the three hormones, followed by *ZmVDAC1b*, suggesting a key role of both transcripts under specific environmental conditions. GA induction of *ZmVDAC4b* transcripts also suggests a particular function of this hormone level, though with these data, it is not possible to correlate it to *Fusarium*-induced GA increased levels. Upregulation of *VDAC* genes by SA was expected, as *Fusarium* infection apparently promoted activation of the SA signaling pathway and upregulation of most *ZmVDAC* genes (Figure 8). Interestingly, transcript levels of *ZmVDAC2* and *ZmVDAC3b* did not show an induction with the three hormones tested, despite a slight increase observed during *F. verticilloides* infection, supporting the hypothesis of nonredundancy between isoforms and suggesting a more specific role for these two genes during *F. verticillioides* invasion. In the case of MeJa treatment, so far, there are no experimental reports on the effect of JA on plant VDAC transcripts. Results show a low transcript level for *ZmVDAC1a*, *ZmVDAC2*, *ZmVDAC3b*, and *ZmVDAC4a* that may be produced by the limited growing of seedlings; however, *ZmVDAC1b* and *ZmVDAC4b* significantly increased their levels, despite growth retardation (Figure 8c,d). This may suggest exclusive functions for these two members. Based on previous experiments, we have hypothesized that MeJa can induce ROS production and alterations in mitochondria dynamics that eventually lead to cell death [48]; hence, *ZmVDAC1b* and *ZmVDAC4b* may have a crucial role during these signaling events.

## 4. Materials and Methods

### 4.1. Identification and Analysis of VDAC Family Members in Maize Genome

Gene sequences of maize VDACs were obtained from UniProt [21], NCBI [22], ENA [23], Ensembl Plants [24], and Maize GDB [25] databases. All identified sequences were submitted to the InterPro database [49] to confirm the Porin3 domain. ProtParam tool, provided by Expasy [50], was used to determine the molecular weight (Mw). BioEdit Sequence Alignment Editor software [51] was used to align and compare nucleotide and amino acid VDAC sequences from maize and plant orthologues. AtVDACs amino acid sequences were used as templates to identify the MPS domain.

Gene structure of cds sequences of the *ZmVDAC* family was analyzed with the Gene Structure Display Server 2.0 [52]. A secondary structure was obtained using MINNOU [53]. Alpha Fold 2 [54,55] and PyMOL [56] were used to predict, visualize, and overlap, respectively, three-dimensional structures of ZmVDAC proteins. The crystal structure of MmVDAC1 (PDB: 3emn) was used as a template for overlapping 3D structures [57].

### 4.2. Phylogenetic Analysis

ZmVDAC and orthologues sequences were aligned using MEGA version X [58], using MUSCLE and Cluster UPGMA with the default options. The phylogenetic tree was constructed with the maximum likelihood method, with bootstrap values of 100 replications.

### 4.3. Analysis of Gene Expression Profiles

Expression data of *ZmVDAC* genes were obtained from the MaizeGDB database [17,18,25]. The expression pattern was visualized as a heat map, reflecting absolute quantification values as Log_2_ values. Expression profiles were represented as red, yellow, and green for high, medium, and low levels respectively.

### 4.4. Identification of Cis-Acting Regulatory Elements in Promoter Region of ZmVDAC Genes

Promoter sequences (−2000 bp) of *ZmVDACs* gene sequences were extracted from Ensembl Plants [24], and the *cis*-acting elements were predicted by New PLACE software [59].

### 4.5. Plant Materials and Growth Conditions

*Zea mays* var. VS535 seeds were disinfected with 2% (*v/v*) household bleach for two min with constant stirring. Hypochlorite was removed from seeds by washing extensively with sterile deionized water. After disinfection, seeds were shaken for 1.5 h at 150 rpm and 30 °C [60] in 1 mL/seed of sterile deionized water (control), 1 mL/seed of 20,000 *F. verticilloides* MY3 conidia/seed (infected), 1 mL/seed of 0.5 mM salicylic acid (SA) (Sigma, St. Louis, MO, USA), 1 mL/seed of 0.05 mM methyl jasmonate (MeJa) (Aldrich, St. Louis, MO, USA) or 1 mL/seed of 0.14 mM gibberellic acid (GA) (Sigma, St. Louis, MO, USA). Then, seeds were sown along with their inoculation water or hormone solution on wet filter paper and germinated in the dark and 30 °C for 1, 2, 3, and 6 days, for control and infected conditions, and only for 3 days for hormone treatments. Each experiment was carried out three times, independently.

### 4.6. Measurement of Root and Coleoptile Lengths

Primary roots and coleoptile lengths, and several secondary roots were measured for control and infected seedlings at different germination times. The experiment was replicated three times using 10 seedlings per replica.

### 4.7. Electrolyte Leakage

Seedlings at each imbibition time were submerged on 0.02% *v*/*v* tween 20 (Sigma, St. Louis, MO, USA). The solution was stirred during conductivity determination. The data were collected every 10 min for 1 h using a digital conductivity meter (HANNA Instruments). Results are representative of three independent experiments and presented as µSmin^−1^cm^−1^dry weight^−1^.

### 4.8. Determination of H_2_O_2_ in Maize Seedlings

The content of H_2_O_2_ in control and infected maize seedlings at different times of germination was determined as described by [61]. Briefly, seedlings were ground in liquid nitrogen. The resulting fine powder was mixed with 1 mL of 25 mM Tris-HCl pH 7.5 and centrifuged at 4000 rpm for 1 min, and 100 µL of the supernatant was mixed with xylenol orange (Sigma-Aldrich, St. Louis, MO, USA) solution prepared at the time of use. Immediately, the samples were mixed in a vortex and incubated for 30 min at 30 °C. H_2_O_2_ was quantified by monitoring the absorbance at 560 nm and compared with a standard curve made with 0.5 to 6 nmol H_2_O_2_. Results are presented as the content of H_2_O_2_ per dry weight gram and the reported data are the mean of three independent experiments with three technical replicates.

### 4.9. Evan’s Bue Staining

To visualize cell death, 6 days of coleoptiles were subjected to Evan’s blue staining. Control and infected cotyledons were soaked in Evan’s blue solution (0.25 g Evan’s blue in 100 mL of 0.1 M CaCl_2_ solution at pH 5.6) for 1 to 2 min, distained with plenty of water and observed under a light microscope (Olympus CH30).

### 4.10. RNA Extraction and cDNA Synthesis

RNA was obtained from non-imbibed seeds and coleoptiles using TRIzol (Invitrogen, Carlsbad, CA, USA) according to the manufacturer’s instructions. Total RNA was quantified on a BioDrop µLite+ (Biochrom, Cambridge, UK) and integrity was evaluated on a 2% agarose gel. RNA samples underwent DNase treatment using RQ1 RNase-Free DNase System from Promega (Madison, WI, USA), following the protocol provided by the manufacturer.

For cDNA synthesis, 1 µg of RNA was mixed with 1 µg of 20 µM oligo dT and nuclease-free water to complete 10 µL of the reaction mixture. The cDNA was denatured at 70 °C for 5 min and chilled on ice for 5 min. ImProm-II Reverse Transcription System from Promega (Madison, WI, USA) was used to obtain cDNA with the following reaction conditions: 1 cycle of annealing at 25 °C for 5 min, extension at 42 °C for 60 min, and denaturing at 70 °C for 15 min for 1 cycle. cDNA was stored at −20 °C until use.

### 4.11. Determination of ZmVDAC Genes by PCR

To determine the transcript levels of *ZmVDAC3a*, *ZmVDAC5a*, *ZmVDAC5b*, and *Zm18S* (reference gene) from coleoptiles, three independent samples were used. For the reaction, 2 µL cDNA, 0.6 µL forward and reverse primers 10 µM, 12.5 µL PCR Master Mix Promega (Madison, WI, USA), and 9.3 µL nuclease-free water were mixed at 4 °C. PCR conditions for amplification of each gene fragment are specified in Appendix A, and primers are listed on Appendix A. PCR products were resolved on a 2% agarose gel. All primers were designed using cDNA sequences of *Zea mays* B73_Ref_Gen_v4 from the Ensembl Plants database [24] and AmplifX 2.1 software [62] to amplify an exon-exon junction region of each gene. The Basic Local Alignment Search Tool (BLAST) and PRIMER-BLAST alignment analysis tool from the NCBI database [22], were used to assure the specificity of each pair of primers. Finally, to discard homodimer, heterodimer, and secondary structures formation, OligoAnalyzer Tool from Integrated DNA Technologies (IDT) [63] was used.

### 4.12. Transcript Level Profiles of ZmVDAC Genes by qPCR

Quantitative PCR (qPCR) in a Mic qPCR Cycler (Bio Molecular Systems) was performed to analyze *ZmVDAC* transcript levels. The reaction mixture contained 10 µL Syber Green Master Mix from Life Technologies (Carlsbad, CA, USA), 0.3 µL forward and reverse primers 10 µM, 2 µL cDNA, and 7.4 µL nuclease-free water. Reaction conditions were as follows: a holding state at 50 °C for 2 min; a second holding state at 95 °C for 10 min; a cycling stage of 40 cycles at 95 °C for 15 s and 60 °C for 60 s; and finally a melting curve at 95 °C for 15 s and 60 °C for 60 s. Primers are listed in Appendix A. The relative gene expression levels were calculated using the 2^−ΔΔCt^ method [64] and previously validated primers for *the Zm18S* gene [65], as the internal reference. Three biological samples with two technical replicates were used for qPCR analyses.

### 4.13. Statistical Analysis

All histograms and graphics were made using Microsoft Excel 365. GraphPad Prism 9.4.1 was used to process the data. One- or two-way analysis of variance was performed, following a Tukey test when necessary to determine the significance of the differences between means. Error bars indicate standard deviation (SD), and significance differences using *p* < 0.05 are indicated with *. Significant differences between groups are indicated with different letters (a–d).

## 5. Conclusions

In this study, the *ZmVDAC* family was identified and described. Nine members were proposed as bona fide VDAC genes with a high potential to encode functional VDAC proteins. Each member of the *ZmVDAC* family was characterized in silico, and nomenclature was proposed according to phylogenetic relationships as follows: *ZmVDAC1a*, *ZmVDAC1b*, *ZmVDAC2*, *ZmVDAC3a*, *ZmVDAC3b*, *ZmVDAC4a*, *ZmVDAC4b*, *ZmVDAC5a*, and *ZmVDAC5b*. Through bioinformatic analysis, it was found that the promoter region of all the ZmVDAC genes has biotic- and hormonal-responding sequences. The expression profile of Fusarium-infected coleoptiles showed that ZmPR1 and ZmPDF1 were upregulated during the early stages of infection while each of the ZmVDAC genes were upregulated on day 6. Transcription profile of the ZmVDAC genes during hormone treatment showed that ZmVDAC1b and ZmVDAC4b were upregulated in each one of the three hormone treatments, though the levels of transcription of the remaining ZmVDACs genes differed in each of the hormone treatments, suggesting no redundancy among ZmVDACs.

Additionally, the results obtained in this study are relevant to a better understanding of the molecular bases of maize responses during the early stages of *Fusarium verticillioides* invasion.

## Figures and Tables

**Figure 1 plants-12-02542-f001:**
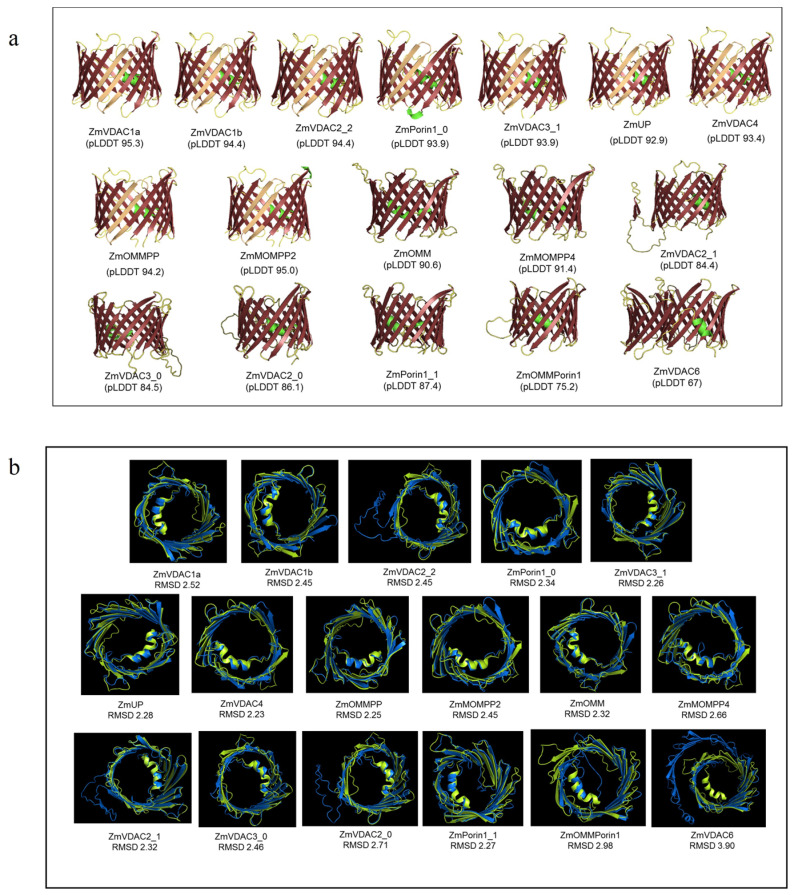
Structural features of the 17 putative ZmVDAC proteins. (**a**) Three-dimensional structure of ZmVDAC proteins represented by a ribbon diagram was predicted using AlphaFold 2.0. All the proteins, except ZmVDAC6, showed the α-helix at the N-terminal and the typical β-barrel conformation; however, only 11 of the proteins have demonstrated high reliability in the structure prediction, as indicated by the pLDDT values > 90. (**b**) Superposition of ZmVDAC 3-D protein models (blue) and MmVDAC1 (PDB:3emn) (green) 3D structure was performed using PyMOL. The structural similarity was found for 13 VDAC proteins, as indicated by the RSMD values similar to or lower than the values of the superimposed structures of AtVDAC1 and MmVDAC1 (2.49). Considering both values, pLDDT and RMSD, only 10 maize gene sequences had a high probability to produce a functional VDAC protein.

**Figure 2 plants-12-02542-f002:**
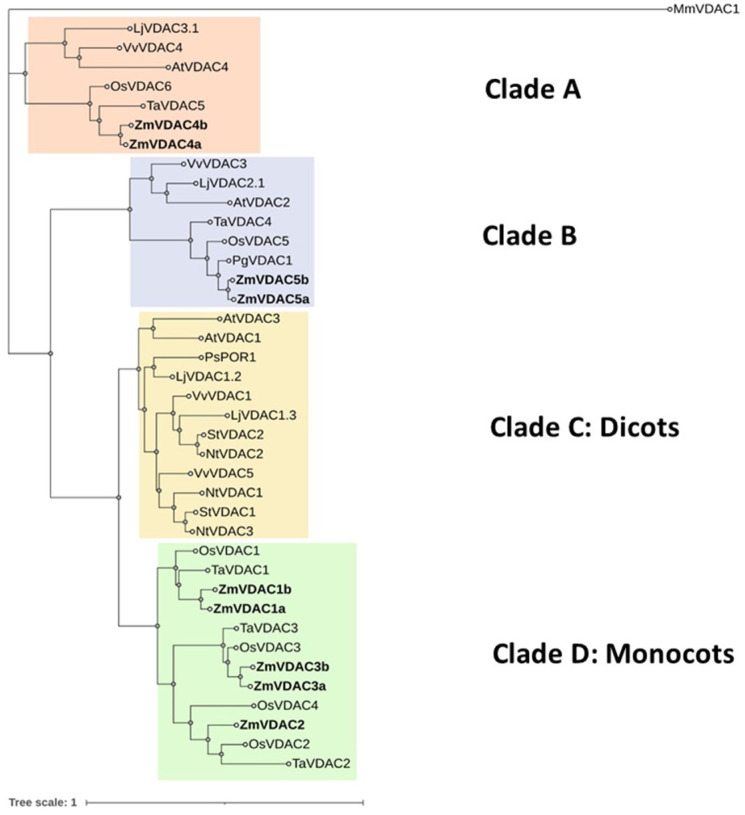
Phylogenetic relationship between VDAC proteins from maize and their orthologues in wheat, rice, pearl millet, tobacco, Arabidopsis, grape, *Lotus japonicus*, pea, and potato. The phylogenetic tree was constructed in MEGA X using the maximum likelihood method (100 bootstraps). VDAC families are separated into four major clades: A and B have members from monocots and dicots, whereas C and D have specific members from dicots and monocots, respectively. The nine VDAC putative members of the *Zea mays* family are distributed in A, B, and D clades.

**Figure 3 plants-12-02542-f003:**
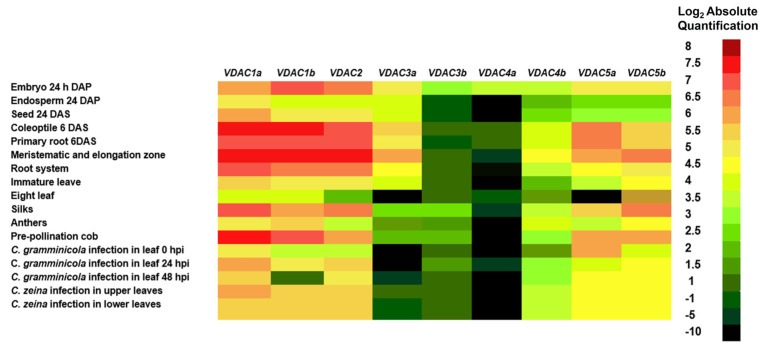
Tissue-specific transcript level profile of the *ZmVDAC* family as determined by Stelpflug et al. [17], Hoopes et al. [18], and Winter et al. [28]. DAP: days after pollination. DAS: days after sowing. The nine *ZmVDAC* sequences chosen are transcribed in different developmental stages and during the plant infection by *Colletotricum graminicola* and *Cercospora zeina*.

**Figure 4 plants-12-02542-f004:**
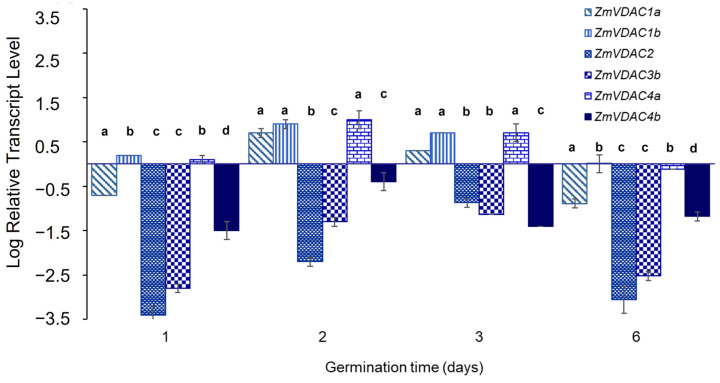
Transcript levels of *ZmVDAC1a*, *ZmVDAC1b*, *ZmVDAC2*, *ZmVDAC3b*, *ZmVDAC4a*, and *ZmVDAC4b* in coleoptiles on days 1, 2, 3, and 6 (d) of germination. Error bars indicate the SD (n = 6) from three independent biological replicates. Different letters (**a**–**d**) within the same time represent significant differences at *p* < 0.05. Six of the nine *ZmVDAC* family members were specifically amplified by qPCR, and their transcript profile varied along the 6 days of germination.

**Figure 5 plants-12-02542-f005:**
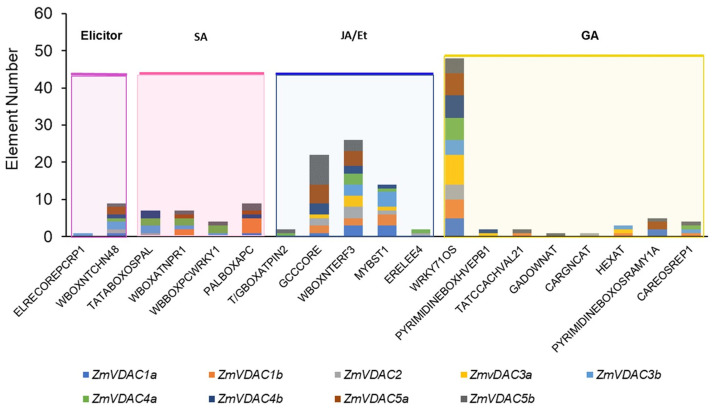
Regulatory elements found on the nine *ZmVDAC* family members 2000 bp upstream of the promoter region. Predictions were made using New PLACE software. *Cis* regulatory elements for elicitor responses, salicylic acid (SA), jasmonic acid/ethylene (JA/Et), and gibberellic acid (GA) were found in most of the *ZmVDAC* family genes. The major number of elements were found in JA/Et and GA.

**Figure 6 plants-12-02542-f006:**
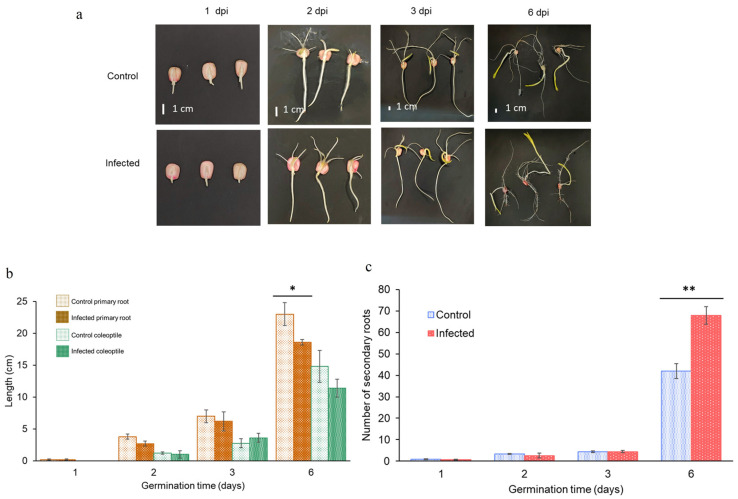
Morphological effect of *F. verticillioides* infection on maize seedling development and quantitative determination of control and infected seedling morphology during 6 days post-infection (dpi). (**a**) Visual effects of *F. verticillioides* infection on seedling growth at 1, 2, 3, and 6 d of germination. (**b**) Primary root (brown) and coleoptile (green) length (cm) (**c**) Number of control (blue) and infected (red) secondary roots. Measurements were made using 10 seedlings for each determination. The final data are the result of three biological and independent replicates. Asterisks “*” indicates significant differences between treatments at *p* < 0.05 (“**”, *p* < 0.01). No morphological changes were observed in the seedlings infected with *F. verticillioides*, except for the shorter primary roots and higher number of lateral roots in the infected seedlings than in the control ones on the 6the day of germination.

**Figure 7 plants-12-02542-f007:**
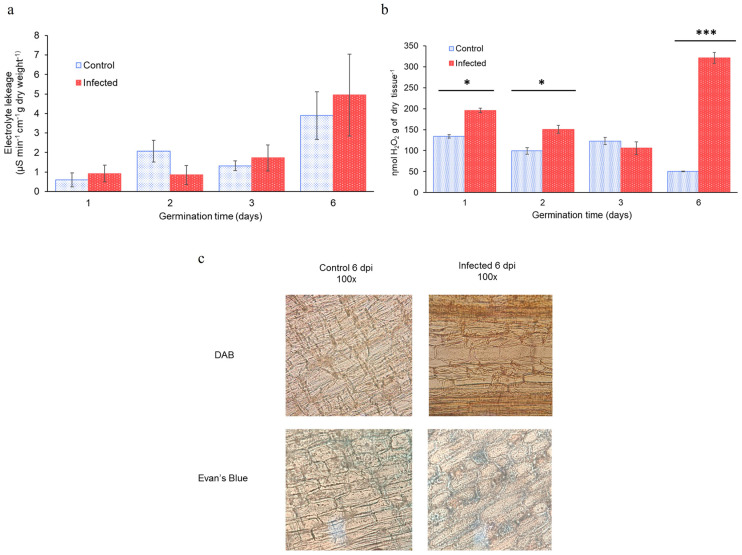
Seedling’ physiological response to *F. verticillioides* infection during 6 dpi. Control (blue) and infected (red) seedlings are shown. (**a**) Electrolyte leakage as a determination of membrane integrity (µS min^−1^cm^−1^ g of dry weight^−1^) was not different between control and *F. verticillioides*-infected seedlings. (**b**) ROS was quantified to determine H_2_O_2_ production (ηmol H_2_O_2_ g of dry tissue^−1^), and it was significantly different on days 1 and 6 of *F. verticillioides* infection. Determinations were made using three seedlings. The final data are the result of three technical replicates, with three biological replicates. Asterisks “*” represent significant differences between treatments with *p* < 0.05 (“***”, *p* < 0.001). (**c**) Micrographs of cotyledons stained with DAB (**up**) and Evan’s blue (**down**) 6 days after inoculation with *F. verticillioides* show no difference with the control seedlings.

**Figure 8 plants-12-02542-f008:**
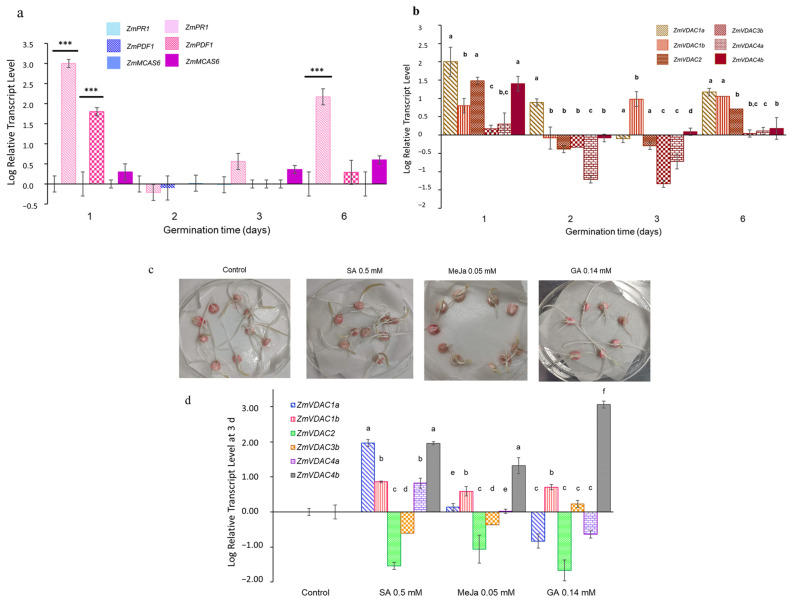
Effect of the infection of *F. verticillioides* and hormones on plant growth and defense genes and *ZmVDAC* transcript levels. (**a**) Transcript levels of *ZmPR1*, *ZmPDF1*, and *ZmMCAS6* in coleoptiles during 6 dpi with *F. verticillioides* MY3 were higher on 1 and 6 dpi. cDNA from uninfected coleoptiles were used as the control, and the *18S* gene was used as the reference gene. Blue bars, control condition; pink bars, infected condition. Error bars indicate the SD (n = 6) from three independent biological replicates. Asterisks “***” indicate significant differences between treatments at *p* < 0.001. (**b**) Transcript level profile of *ZmVDAC* genes in coleoptiles during 6 dpi with *F. verticillioides* MY3 were different from the control coleoptiles, and most of them were more expressed on 1 dpi. Error bars indicate the SD (n = 6) from three independent biological replicates. Different letters (a–d) within the same time represent significant differences at *p* < 0.05. (**c**) Hormone treatments of maize seed affected the seedlings growth differently: enlargement of the coleoptile was observed with SA, reduction of overall growth with JA, and increase in the root architecture with GA. Seedlings are shown at 3 days of growth at 30 °C in darkness. (**d**) Transcript levels of six *ZmVDAC* genes in coleoptiles at 3 days after treatment were affected by the addition of 0.5 mM salicylic acid (SA), 0.05 mM methyl jasmonate (MeJa), and 0.14 mM gibberellic acid (GA). Error bars indicate the SD (n = 6) from three independent biological replicates. Different letters (a–f) represent significant differences between treatments at *p* < 0.05.

**Table 1 plants-12-02542-t001:** Proposed nomenclature for the nine maize VDAC sequences with the highest probability to encode functional proteins.

Maize Sequence Original Name	Closest Orthologue	Assigned Name	Nucleotide Sequence Length (bp)	Amino Acid Sequence	Molecular Weight (kDa)	UniProt ID	NCBI	Ensembl Plants	Maize GDB
ZmVDAC1a	OsVDAC1 TaVDAC1	ZmVDAC1a	831	276	29.84	Q9SPD9	NP_001105619.1	Zm00001e098580	Zm00001d005649
ZmVDAC1b		ZmVDAC1b	831	276	29.65	A0A1D6I2X 6	NP_001104948.2	Zm00001eb310830	Zm00001d020219
ZmVDAC2_2	OsVDAC2	ZmVDAC2	831	276	29.18	Q9SPD7	NP_001104949.1	Zm00001eb294130	Zm00001d038840
ZmPorin1_0	OsVDAC3 TaVDAC3	ZmVDAC3a	828	275	29.78	K7VJ77	NP_001278544.2	Zm00001eb357390	Zm00001d011242
ZmVDAC3_1		ZmVDAC3b	828	275	29.79	B4FX24	NP_001141587.1	Zm00001eb160640	Zm00001d044318
ZmUP	AtVDAC4	ZmVDAC4a	831	276	29.62	A0A804LDA0	NM_001154290.2	Zm00001eb003070	Zm00001d027578
ZmVDAC4		ZmVDAC4b	831	276	29.59	B6T1E3	NM_001155629.2	Zm00001eb003070	Zm00001d048362
ZmOMMPP	OsVDAC5	ZmVDAC5a	828	275	29.23	A0A1D6JS28	NM_001154014.2	Zm00001eb007730	Zm00001d028102
ZmMOMPP2		ZmVDAC5b	828	275	29.18	C4IYM7	NP_001288497.1	Zm00001eb400750	Zm00001d048149

## Data Availability

The sequences analyzed during the current study are available in the UniProt (https://www.uniprot.org/, accessed on 22 May 2023), NCBI (http://ncbi.nlm.nih.gov, accessed on 22 May 2023), Ensemble Plants (https://plants.ensembl.org/index.html, accessed on 22 May 2023), ENA (https://www.ebi.ac.uk/ena/browser/home, accessed on 22 May 2023), and MaizeGDB (https://www.maizegdb.org/, accessed on 22 May 2023) repositories. Data supporting results can be found in the Appendix A attached to this manuscript.

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
