# Peer review of "Identification and Characterization of VDAC Family in Maize"

_plants, 2023, doi:10.3390/plants12132542_

Round 1

Reviewer 1 Report

The manuscript was interesting and overall it is good quality. There are a few minor English language items that need to be addressed.

Line 234: Please change "de" to "the". 

Also, use spell check to check for additional mistakes that I may have missed.

Author Response

We appreciatte the comments of the reviewer and we made the changes were needed, including the line 234 were was mispealled "the".

Reviewer 2 Report

The manuscript submitted by Rodriguez-Saavedra et al. concerns the characterization of the VDAC family in maize for the first time.

The research design is accurate and appropriate to the aim of the manuscript, but the sections have not been written correctly, arising some confusion.

Eliminate "maize" from the keywords and add a new one that is not present in the title

In line 55 you talk about V. piasezkii VDACs but did not mention them in the list before. Please add them.

The aims described in the introduction need to be reformulated and shorten. I suggest to eliminate "was identified from databases" in line 80 and to reformulate lines 83-91 as follows "The ZmVDAC family response to F. verticillium infection was investigated together with the analyses of GA, SA and MeJa resposne as signaling hormones of biotic stress response."

In the results you are often rewriting Methods that are not useful for data understanding or explaination. For example, lines181-182 says "using the maximum likelihood method with MEGA X", but it is already reported in lines 653-655. Please check all results and eliminate the informations about Methods already reported in teh dedicated section.

Lines 211-213: Eliminate from "maize..." to "...[17,18,28,33]" whicha re also exactly the same citations reported in lines 657-658.

must be checked by a native speaker because some sentences are very long and difficult to udnerstand

Author Response

We appreciatte the suggestions here the responses

Comment 1: The research design is accurate and appropriate to the aim of the manuscript, but the sections have not been written correctly, arising some confusion.

Response 1: We eliminate some of the sentences to make it more fluent.

Comment 2: Eliminate "maize" from the keywords and add a new one that is not present in the title

Response 2: We changed “maize” for “seedling development”

Comment 3: In line 55 you talk about V. piasezkii VDACs but did not mention them in the list before. Please add them.

Response 3: You are right Xu et al. (2021) reported VDAC genes for two grape species: vinifera (susceptible) and piasezkii (resistant). Both species have same number of VDAC genes. We added in page 2 line 50: “Vitis vinifera and Vitis piasezkii have six [10]”

Point 4: The aims described in the introduction need to be reformulated and shorten. I suggest to eliminate "was identified from databases" in line 80 and to reformulate lines 83-91 as follows "The ZmVDAC family response to F. verticillium infection was investigated together with the analyses of GA, SA and MeJa response as signaling hormones of biotic stress response."

Response 4: Thank you for the suggestion. We did the changes and eliminate the non-required information in the last parragraph of the page 2

Point 5: In the results you are often rewriting Methods that are not useful for data understanding or explanation. For example, lines181-182 says "using the maximum likelihood method with MEGA X", but it is already reported in lines 653-655. Please check all results and eliminate the information about Methods already reported in the dedicated section.

Response 5: For clarity we elimminate information about methodology from the results section.

Point 6: Lines 211-213: Eliminate from "maize..." to "...[17,18,28,33]" which are also exactly the same citations reported in lines 657-658.

Response 6: We eliminate those references and rearrarange reference numbering.